# An allosteric inhibitor of the Zika virus NS2B-NS3 protease with oral efficacy in mouse models

Jesus M. Ontoria [1] ✉, Esther Torrente[1], Antonino Missineo[2], Cristina Alli[2], Rita Graziani[2], Silvia Conti[2], Monica Bisbocci[2], Antonio Quotadamo[1], Federica Ferrigno[1], Alessandra Corio[1], Giovanni Ievoli[1], Leda Bencheva[1,4], Jérôme Amaudrut[1], Silvana Vasile[1], Elisa Beghetto[2], Chantal Paolini[2], Nadine Alaimo[2], Maria Veneziano[1], Martina Nibbio[1], Maria V. Orsale[1], Giulia Proto[1], Fabrizio Colaceci[1], Laura Orsatti[1], Vincenzo Pucci[1,5], Romano Di Fabio[1,6], Licia Tomei[2], Christian Montalbetti[1], Alberto Bresciani[2,7], Carlo Toniatti[3] & Giacomo Paonessa[2] ✉

The mosquito-transmitted Zika virus (ZIKV) poses a global health threat, with no approved antiviral drugs or vaccines currently available. Here, we report the discovery of a series of ZIKV NS3 protease inhibitors identified through phenotypic high-throughput screening (HTS) using a ZIKV replicon-based cellular assay, and the subsequent selection of resistant mutants. These inhibitors, characterized by the presence of an *N*-acylsydnone imine group, bind to a previously undescribed allosteric pocket of the protease, locking the enzyme into a catalytically inactive conformation. We describe the characterization of IRBM-Z-1, our initial allosteric hit and IRBM-Z-2, a potent inhibitor of ZIKV infectivity and other orthoflavivirus proteases with a favourable in vitro and in vivo ADME profile, resulting in oral efficacy against ZIKV infection in mouse models, with potential as a prophylactic agent for human use.

Zika virus (ZIKV) is an arthropod-borne, single-stranded RNA virus of the *Flaviviridae* family, which also includes viruses such as Dengue (DENV), Japanese encephalitis (JEV), and West Nile (WNV)[1]. ZIKV is primarily transmitted by the bite of infected *Aedes aegypti* mosquitos but can also spread via sexual contact[2]. While most infections are asymptomatic, approximately 20% of infected individuals develop clinical manifestations, including thrombocytopenia, ocular damage, multi-organ failure, and, notably, congenital abnormalities such as microcephaly in foetuses and newborns[1]. In adults, ZIKV has been linked to severe neurological disorders, including meningitis, seizures, myelitis, encephalitis, and Guillain–Barre syndrome[3]. Following a 2015 outbreak in Brazil, ZIKV spread rapidly across South American

countries reaching epidemic levels[4]. Despite its global impact, there are currently no approved vaccines or antiviral therapies for the prevention and treatment of ZIKV infection, representing an unmet medical need. Development of ZIKV antivirals is still at an early stage, with several experimental compounds reported to target different aspects of the viral lifecycle. These include SBI-0090799, which inhibits replication compartment formation; SYC-1307, an allosteric inhibitor of the NS2B-NS3 protease; and Galidesivir, a C-nucleoside analogue, that inhibits the viral RNA-dependent RNA polymerase (RdRp)[5]. The discovery of safe and effective antiviral agents against ZIKV therefore remains a research priority for pandemic preparedness and could play a significant role in mitigating the global burden of ZIKV

[1]Department of Drug Discovery, IRBM S.p.A., Pomezia, Italy. [2]Department of Biology and Translational Research, IRBM S.p.A., Pomezia, Italy. [3]CSO, IRBM S.p.A., Pomezia, Italy. [4]Present address: Sibylla Biotech S.p.A., Bresso, Italy. [5]Present address: Johnson & Johnson, Beerse, Belgium. [6]Present address: Drug Discovery Unit, Vita Salute San Raffaele University, Milan, Italy. [7]Present address: Tycho S.r.l., Bergamo, Italy. ✉e-mail: j.ontoria@irbm.com; g.paonessa@irbm.com

infections[6]. In recent years, the use of reverse genetics and the establishment of stable cell lines harbouring viral replicons have facilitated antiviral drug discovery by enabling phenotypic screening in biologically relevant contexts[7]. Here, we report the development of a ZIKV replicon-based cellular assay, which we use in a high-throughput screening (HTS) campaign to identify small-molecule inhibitors of ZIKV replication. Target deconvolution through resistance selection and enzymatic validation reveal a single compound class as ZIKV NS3 protease inhibitors. Further optimization yields orally bioavailable inhibitors with favourable pharmacokinetic properties and potent in vivo efficacy in a mouse model of ZIKV infection.

## Results

### HTS and the identification of IRBM-Z-1 as a non-competitive allosteric inhibitor of ZIKV NS3 protease

To identify ZIKV replication inhibitors, we engineered a ZIKV replicon (Fig. 1a) in which the structural genes were deleted and NanoLuciferase and Neomycin-resistant coding sequences were inserted into the genome downstream of an IRES element. Upon transfection of this ZIKV replicon RNA into Vero cells, followed by antibiotic selection, we established a stable cell line in which luciferase expression was strictly proportional to replicon replication. Using this stable cell line, we established a phenotypic-based assay and screened over 120,000 compounds from the CNCCS (Italian National Compound and Screening Collection)[8] library to directly identify molecules that inhibit viral replication, with minimal to no effect on cellular viability. Hits were prioritized via a screening funnel (Fig. 1b) that included target deconvolution. This approach led to the identification of compounds with micromolar potency in the ZIKV replicon assay ($EC_{50} < 10\,\mu M$), no cytotoxicity up to 32 μM concentration, and attractive chemical structures. Eight of the most potent hits, each representing distinct chemotypes, were used to generate compound-resistant cellular clones by culturing cells with a 5-fold higher concentration of each compound's respective $EC_{50}$ potency. We successfully isolated and expanded several compound-resistant cellular clones for one of these compounds, IRBM-Z-1 (Fig. 1b), all of which harboured an ATC → ACC codon change in amino acid position 156 in NS3, resulting in an isoleucine → threonine substitution (I156T). Remarkably, the I156T mutation in NS3 was the only one consistently associated with resistance in all selected clones (for a representative sequence see Supplementary Fig. 3).

To further elucidate the mechanism of inhibition for IRBM-Z-1 and confirm NS3 as the real target, we developed an in vitro enzymatic assay using recombinant wild type (wt) and I156T mutant NS2B-NS3 protease. In parallel, to avoid any possible cellular interference from the selected clone, we established a novel stable cell line harbouring an NS3 I156T mutated replicon. As shown in Table 1, IRBM-Z-1 proved to be a low micromolar in vitro inhibitor of the wt NS2B-NS3 protease with an $IC_{50}$ of 1.8 μM. This result, in line with the replicon potency of 6 μM (Fig. 1c) and ZIKV antiinfectivity activity in cell culture assays ($EC_{50}$: 6.25 μM), confirms that NS3 protease is the primary biological target. As expected, the introduction of the I156T mutation in NS3, strongly reduced the inhibitory effect of IRBM-Z-1, in both the enzymatic and replicon assays (Table 1 and Fig. 1d). In contrast, CN-716, a known covalent inhibitor of the NS3 catalytic site[9], retained activity against both wt and mutant proteases but showed no effect in the cell-based ZIKV replicon assay due to its poor permeability.

Notably, the NS3 proteases of all four DENV serotypes, despite their high sequence homology with ZIKV, naturally contain a threonine at position 156[10]. To investigate the role of this residue in compound sensitivity, we evaluated IRBM-Z-1 against both wt DENV2 NS3 and T156I mutant proteases in an in vitro system, in which the isoleucine residue found in the ZIKV was reintroduced. IRBM-Z-1 showed no inhibitory activity against wt DENV2 protease but gained activity on the T156I mutant with an $IC_{50}$ of $3.9 \pm 0.4\,\mu M$ comparable to the wt ZIKV

protease (Table 1 and Fig. 1d). These results strongly suggest that the amino acid in position 156 plays a critical role in determining resistance or sensitivity to IRBM-Z-1. The interaction between IRBM-Z-1 and the NS3 protease was further characterized through binding kinetics and mechanism of action studies[11]. IRBM-Z-1 exhibited a dissociation constant ($K_D$) of $1.52 \pm 0.75\,\mu M$, in line with the biochemical $IC_{50}$ value ($1.8 \pm 0.8\,\mu M$) (Fig. 1e). Importantly, IRBM-Z-1 retained its potency across increasing substrate concentrations relative to $K_M$, indicating a non-competitive mode of inhibition. Thus, IRBM-Z-1 can be classified as a non-competitive allosteric ZIKV NS3 protease inhibitor (Fig. 1e, f). This finding highlights an opportunity to explore a novel mechanism of action that might overcome the challenges associated with developing potent orthosteric Zika protease inhibitors with effective cellular activity[12,13].

### Crystal structure of the NS2B-NS3 protease in complex with R-(+)-IRBM-Z-1

Crystal structures of the ZIKV NS2B-NS3 protease available in the Protein Data Bank (PDB)[14] reveal two major conformational states[15]. In the first, or *closed* conformation (PDB ID 5GPI[16], Fig. 2a), the NS2B cofactor (residues 67–87) folds over the NS3 domain, stabilizing the catalytic triad in a fully formed and active site. The C-terminal segment of NS3 (residues 153–170) adopts a β-sheet conformation that contributes to this active architecture. In contrast, the second or *open* conformation (PDB ID 5GXJ, Fig. 2b) shows the C-terminal region of NS3 forming a turn, preventing NS2B from achieving the closed state and rendering the protease catalytically inactive. We determined the crystal structure of the ZIKV NS2B-NS3 protease in complex with the enantiomerically pure R-(+)-IRBM-Z-1 (PDB ID 9TPG, Fig. 2e and Supplementary Fig. 10), which was five-fold more potent in the biochemical assay compared to the other enantiomer, and showed a two-fold higher activity in the ZIKV replicon assay compared to the racemic mixture (Table 1). In this structure, the ligand is deeply buried into a previously uncharacterized allosteric pocket located far from the catalytic site and induces the protein to adopt an open-like conformation. An additional ligand molecule is bound at the crystal interface close to the symmetry axis and is therefore a crystallographic artifact without biological relevance (Fig. 2d, e, transparent orange sticks). To the best of our knowledge, the binding pocket occupied by R-(+)-IRBM-Z-1 has not been described in prior structural studies, and it is collapsed in the open apo-protein structure (PDB ID 5GXJ), with the C-terminal segment of NS3 partially overlapping with the binding site of the trifluoromethylphenyl moiety (Fig. 2c). Furthermore, this specific site is absent from known X-Ray crystallography and in silico models of allosteric protease inhibitors across other orthoflaviviruses (Supplementary Table 2 and Supplementary Figs. 7–9)[14,15]. A study of 2019 by Yao et al. reports the crystal structure of a ZIKV NS2B-NS3 protease allosteric inhibitor in complex with the DENV2 protease[16]. Although the pocket identified in Yao's study partially overlaps with the allosteric pocket described herein, the ligand engages substantially different interactions with the protease (Supplementary Fig. 7). Detailed analysis of the binding interface reveals several stabilizing interactions between IRBM-Z-1 and the protease (Fig. 2e). First, the N-acylsydnone imine core forms a unique hydrogen bond (H-bond) network with the backbone of residues V126, L149 and G151 involving two H-bond acceptors and one H-bond donor interactions. One of the N atoms of the urea group acts as an H-bond acceptor, stabilizing this unusual tautomeric form of the N-acylsydnone imine mesoionic ring. Second, a π-stacking interaction of the electron deficient sydnone ring with the Y150 side chain is observed, with the phenethyl N-substituent oriented towards the solvent. Finally, the trifluoromethylphenyl group occupies a highly hydrophobic pocket (composed of residues W89, V95, P113, V126, I139 and I147) which is otherwise occupied by residue I156 when the protein adopts the active closed conformation (PDB ID 5GPI, Fig. 2a)[17]. A docking study of the (S)-enantiomer (Supplementary Fig. 6)

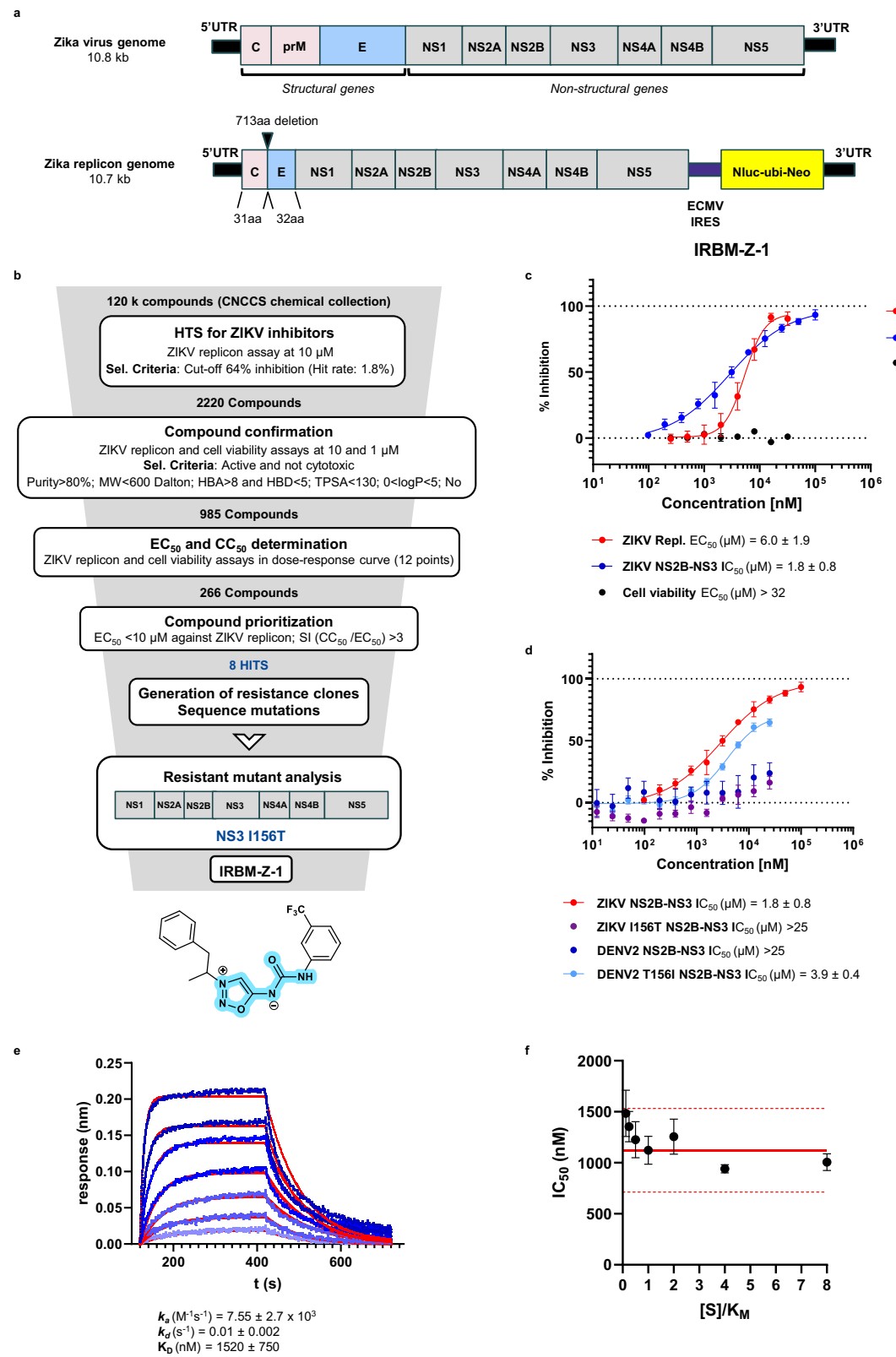

shows that the methyl group is predicted to clash with the NS3 loop residues L128-D129, possibly explaining the enantiomer's lower affinity for the protease (Table 1).

## Structural insights into the I156T resistance mechanism

Comparison of the closed conformation of the ZIKV and DENV2 proteases (PDB ID 5GPI and 2FOM respectively, Supplementary Fig. 8a, b) provides a structural rationale for the resistance mechanism conferred by the I156T mutation observed in ZIKV and the lack of activity of IRBM-Z-1 against the wt DENV2 protease. In fact, the T156 residue in DENV2 engages in a H-bond network that stabilizes the N-terminal NS3 segment in a $\beta$-sheet conformation (Supplementary Fig. 8b). This conformation appears incompatible with IRBM-Z-1 binding, likely contributing to the compound's inactivity against the wt DENV2

**Fig. 1 | Summary of the HTS campaign. a** Schematic representation of the wt ZIKV genome and the engineered replicon used to generate stable cell lines for the phenotypic screen. NanoLuciferase (NanoLuc), Ubiquitin (UBI) and neomycin phosphotransferase resistance (NEO). **b** High-throughput screening (HTS) funnel (Supplementary Figs. 1 and 2); phenotypic screening using ZIKV replicon and cell viability (CellTiter-Glo) assays, generation of compound-resistant replicon cell lines, characterization of resistant clones and identification of IRBM-Z-1 as an NS3 protease inhibitor. Replicon assays were carried out using Vero cells. IRBM-Z-1 chemical structure; the *N*-acylsydnone imine core highlighted in blue. **c** Dose−response curves and $IC_{50}$ and $EC_{50}$ values of IRBM-Z-1 in the ZIKV replicon assay (red curve), NS2B-NS3 protease enzymatic assay (blue curve), and Vero cell proliferation assay (black curve). **d** Dose−response curves and $IC_{50}$ values of IRBM-Z-1 against ZIKV and DENV2 NS2B-NS3 protease activity, including wt and mutants with substitutions at residue 156. In both panels, percent inhibition of enzymatic activity is plotted against compound concentration (nM). $EC_{50}$ values were determined using Prism software. All data represent the mean ± SD from at least three independent experiments. **e** Representative BLI (biolayer interferometry) sensorgrams (blue curves) and 1:1 fitting (red curves) showing the IRBM-Z-1 and ZIKV NS2B-NS3 protease interaction. Binding parameters represent the mean ± SD of six independent experiments. **f** Mechanism of action of IRBM-Z-1 on the NS2B-NS3 protease. The solid red line represents the $IC_{50}$ value at substrate concentration $[S] = K_M$, dashed red lines indicate the confidence interval ($IC_{50}$ $[S] = K_M ± 3$ fold SD). $IC_{50}$ values are shown in Supplementary Fig. 12. All data represent the mean ± SD from at three independent experiments.

## Table 1 | Biochemical and cellular potencies against ZIKV, DENV2 and WNV NS3 proteases and replicons

| Biological assay | CN-716 | IRBM-Z-1 | *R*-IRBM-Z-1 | S-IRBM-Z-1 | IRBM-Z-2 |
|---|---|---|---|---|---|
| ZIKV NS2B-NS3 | 0.5 ± 0.2 | 1.8 ± 0.8 | 0.8 ± 0.2 | 4.6 ± 3.4 | 0.04 ± 0.01 |
| ZIKV I156T NS2B-NS3 | 0.3 ± 0.03 | >25 | >25 | >25 | 3.1 ± 0.2 |
| ZIKV replicon | >32 | 6.0 ± 1.9 | 3.2 ± 0.6 | >32 | 0.03 ± 0.01 |
| ZIKV I156T replicon | N.D. | >25 | >25 | >25 | 0.79 ± 0.04 |
| DENV2 NS2B-NS3 | N.D. | >64 | >64 | >64 | 2.1 ± 0.4 |
| DENV2 T156I NS2B-NS3 | N.D. | 3.9 ± 0.4 | 1.8 ± 0.2 | >25 | 0.4 ± 0.02 |
| DENV2 replicon | N.D. | N.D. | N.D. | N.D. | >25 |
| WNV NS2B-NS3 | N.D. | 4.7 ± 1.2 | 2.5 ± 0.5 | >25 | 0.09 ± 0.02 |
| WNV replicon | N.D. | N.D. | N.D. | N.D. | 0.31 ± 0.16 |
| Cell viability | >32 | >32 | >32 | >32 | >32 |

Replicon assays were performed in Vero cells. Biochemical assays data are expressed as $IC_{50}$ values in μM concentration. Data from the cell-based assays are expressed as $EC_{50}$ values in μM. Cell viability assays are expressed as $CC_{50}$ values in μM. All data represent the mean ± SD from at least three independent experiments except for ZIKV I156T NS2B-NS3 inhibition by CN-716. None of the compounds showed cytotoxicity in the CellTiter-Glo (CTG) cell viability assay at a concentration below 32 μM.
*N.D.* not determined.

protease. Introducing the T156I mutation in DENV2 disrupts this H-bond network, potentially enabling the protease to adopt a conformation more permissive to IRBM-Z-1 binding, consistent with the observed gain of activity. Conversely, I156 in the wt ZIKV is surrounded by the same hydrophobic side chains that form the binding pocket of the trifluoromethylphenyl moiety of IRBM-Z-1 (Fig. 2e). Substitution with threonine (I156T) likely perturbs this hydrophobic pocket and alters the conformation of the N-terminal NS3 region, thereby reducing ligand accessibility or stability within the allosteric site. This structural change may prevent IRBM-Z-1 binding while preserving interactions with other regions of the protein.

### IRBM-Z-2 is a potent non-competitive allosteric inhibitor of ZIKV replication with good pharmacokinetic properties

Extensive structure-activity relationship (SAR) exploration around IRBM-Z-1 using a structure-based design driven the structural biology information, led to the identification of IRBM-Z-2 (Fig. 3)[18], a nanomolar inhibitor of the ZIKV NS3 protease. IRBM-Z-2 inhibits both viral replication and infectivity of ZIKV in cultured cells with comparable potency and no detectable cytotoxicity up to 32 μM concentration (Table 1 and Fig. 3d, e). Furthermore, preliminary experiments suggest that IRBM-Z-2 exhibit antiviral activity against additional ZIKV strains, including Padova and H/PF/2013, in human HuH-7 cells with apparent $IC_{50}$ values ranging from 6 to 20 nM (Supplementary Fig. 13). IRBM-Z-2 lost nearly two orders of magnitude of activity when tested against the I156T NS2B-NS3 mutant, in both biochemical and replicon assays, consistent with what was observed for IRBM-Z-1 (Table 1). Kinetic studies confirmed the binding with the NS2B-NS3 protease with a $K_D$ of 21 ± 5 nM (Fig. 3b). The crystal structure of ZIKV in complex with IRBM-Z-2 (PDB ID 9IBY, Fig. 3c and Supplementary Fig. 11) reveals a binding mode similar to IRBM-Z-1, with an additional stabilizing π-stacking interaction between the dimethyl pyrimidine moiety and residue F116,

likely accounting for the enhanced potency. Moreover, IRBM-Z-2 displays inhibitory activity against other orthoflavivirus proteases, such as DENV2 and WNV, with potencies ranging from nanomolar to low micromolar in both biochemical and cell-based assays (Table 1 and Fig. 3e, a sequence alignment of orthoflavivirus NS2B and NS3 proteins is shown in Supplementary Fig. 5), while showing no measurable activity against human proteases (Fig. 3a). IRBM-Z-2 exhibits favourable physicochemical and pharmacokinetic properties including optimal in vitro and in vivo ADME characteristics and a clean off-target safety profile (Fig. 3a; Supplementary Fig. 18 and Supplementary Table 4). Among the favourable ADME features, this lead compound exhibits low clearance, good to optimal volume of distribution and high oral bioavailability in mice (F 90 %). In particular, IRBM-Z-2 displays an excellent unbound fraction value across the three species tested, especially in humans ($f_u$ 3.6 %), supporting its therapeutic potential. Following a single 100 mg/kg dose administered intraperitoneally (IP) or orally (PO) in C57BL/6 mice, the resulting plasma concentrations exceeded the $EC_{99}$ (1.39 μM, unbound) for more than 24 h (IP) and up to 20 h (PO) with no adverse effects observed (Supplementary Fig. 14). When IRBM-Z-2 was dosed daily at 100 mg/kg IP or PO for five consecutive days, high $C_{max}$ values were observed at day 5 in agreement with those observed in the single-dose studies (Supplementary Fig. 15). No evident signs of toxicity were detected after five days. Furthermore, a maximum tolerated dose study was conducted with daily oral administration of up to 300 mg/kg for seven consecutive days, which was well tolerated and showed no clinical signs of toxicity. Interestingly, the lower daily dose of 100 mg/kg achieved higher $C_{max}$ (23.3 μM) and plasma exposure (208 μM·h), maintaining drug levels well above the $EC_{99}$ for at least 20 h (Supplementary Fig. 16). This pharmacokinetic profile supports the feasibility of a twice-daily (BID) dosing regimen to maintain plasma concentrations above the $EC_{99}$ for a full 24-hour period, as confirmed by a single study

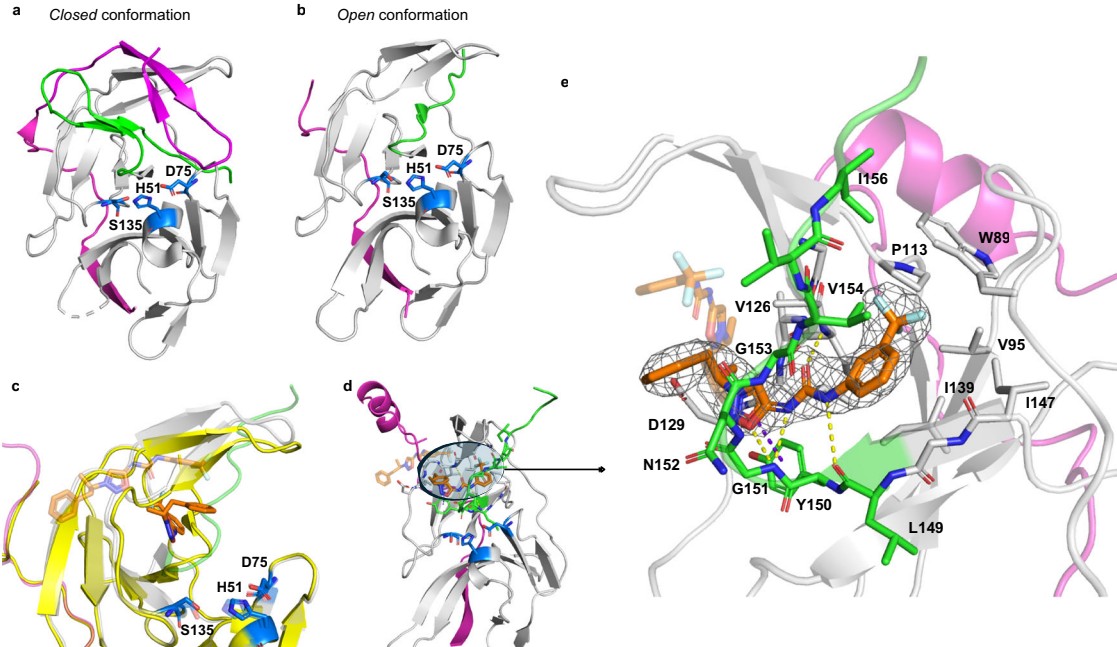

**Fig. 2 | Structural basis of IRBM-Z-1 binding to the ZIKV NS2B-NS3 protease.**
**a** The ZIKV NS2B-NS3 protease in the catalytically active closed conformation, apo structure (PDB ID 5GPI). **b** ZIKV protease in the inactive open conformation (PDB ID 5GXJ). The *C*-terminal segment of NS3 is shown in green, while NS2B is in pink. The residues of the catalytic triad (H51-D75-S135) of the NS2B-NS3 serine protease are displayed as blue sticks. **c** Superposition of the crystal structure of the ZIKV protease in complex with *R*-(+)-IRBM-Z-1 (PDB ID 9TPG, transparent cartoons) and the open conformation (PDB ID 5GXJ, opaque yellow cartoons). The *C*-terminal

segment of NS3 in 5GXJ partially overlaps with the binding pocket of *R*-(+)-IRBM-Z-1. **d** Crystal structure of the ZIKV protease bound to the *R*-(+)-IRBM-Z-1 inhibitor (PDB ID 9TPG), with the allosteric binding site of *R*-(+)-IRBM-Z-1 highlighted. **e** Close-up view of the *R*-(+)-IRBM-Z-1 binding pocket. The ligand is shown in orange sticks. The hydrogen bonds are indicated by yellow dashed lines. The π-stacking interaction between the ligand and the Y150 sidechain is depicted by purple dashed lines. Electron density difference map $2F_O\text{-}F_C$ at +1σ carved at 1.5 Å from the ligand is shown as grey isomesh.

in which animals received 100 mg/kg orally every 12 h (Supplementary Fig. 17). Additionally, an in vitro-in vivo correlation (IVIVC) value of 1 in rats (see Methods) indicates that IRBM-Z-2 primarily undergoes hepatic metabolism, enabling more accurate prediction of human pharmacokinetics and helping streamline the preclinical development process. Based on these findings, we estimated a human oral dose of 506.2 mg once daily to maintain steady-state plasma concentrations above the $EC_{99}$ threshold (133.5 nM, unbound) over a 14-day treatment period[19]. These promising PK and safety data supported the advancement of IRBM-Z-2 into in vivo efficacy studies in a ZIKV infection mouse model.

## Efficacy of IRBM-Z-2 in mouse models of ZIKV infection
The antiviral efficacy of IRBM-Z-2 was next evaluated in an AG129 mouse model of ZIKV infection (Fig. 4a), a well-established system in which viral challenge typically leads to high viral titres in serum, weight loss, lethargy, and hunched posture within a few days of infection[20,21]. In an initial proof-of-concept (PoC) study with a prophylactic approach, mice were challenged IP with ZIKV and immediately treated with IRBM-Z-2 at 100 mg/kg IP, once a day (QD) for five consecutive days (Fig. 4a and Supplementary Table 5). IRBM-Z-2 treatment resulted in rapid and marked antiviral activity, with serum viral RNA levels 3.6 $log_{10}$ lower than those of untreated controls observed by day 1 and 4.5 $log_{10}$ lower levels by the end of the study (Fig. 4b). No adverse clinical signs were observed in the treated group, and by day 5, treated mice were fully protected from ZIKV-induced weight loss compared to the control group (Fig. 4c). Encouraged by this PoC study and considering the favourable drug exposure under the BID regimen (Supplementary Fig. 17), IRBM-Z-2, was further assessed in a 14-day survival and efficacy study, using oral administration (PO). The extended treatment duration of 14 days exceeding the conventional 4–9 days study period, was specifically chosen to enable a comprehensive assessment of IRBM-Z-2

sustained efficacy and to investigate its toxicological profile following prolonged exposure, thereby providing valuable insights into its safety and therapeutic index. Mice were dosed (BID) via oral gavage at 100 mg/kg, starting approximately one hour post-infection and continuing throughout the study (Fig. 4a and Supplementary Table 6).

Even in the 14-day lethal viraemia mouse model, IRBM-Z-2 treatment resulted in serum viral RNA levels more than 6 $log_{10}$ lower than vehicle-treated controls by day 3 post-infection. By day 5, serum viral RNA levels had fallen below the limit of detection and remained undetectable through the end of the study (Fig. 4d). Consistent with earlier results, no significant weight loss or adverse clinical signs of distress were observed in the IRBM-Z-2 treated mice relative to the vehicle-treated group (Fig. 4e). In contrast, all mice in the control group reached the humane endpoint and were euthanized between day 7 and 9, whereas all treated mice survived and recovered fully from the infection (Fig. 4f). These findings highlight the strong potential of IRBM-Z-2 as a drug candidate for the prevention of ZIKA infection.

## Discussion
In the absence of safe and effective vaccines, there remains an urgent need for efficacious antivirals for the treatment of orthoflavivirus infections. For ZIKV, this need is especially critical in pregnant women, the population most at risk[22], where the clinical development of antivirals is severely constrained by ethical and practical challenges underscoring the importance of identifying candidates with a strong safety profile and demonstrated prophylactic efficacy in robust preclinical models. Moreover, prophylactic antiviral use during outbreaks could benefit both local populations and travellers while helping prevent disease spread, as demonstrated by similar strategies employed in malaria control[23]. The discovery of antiviral agents with novel mechanisms of action represents a critical advancement in the ongoing fight against viral diseases.

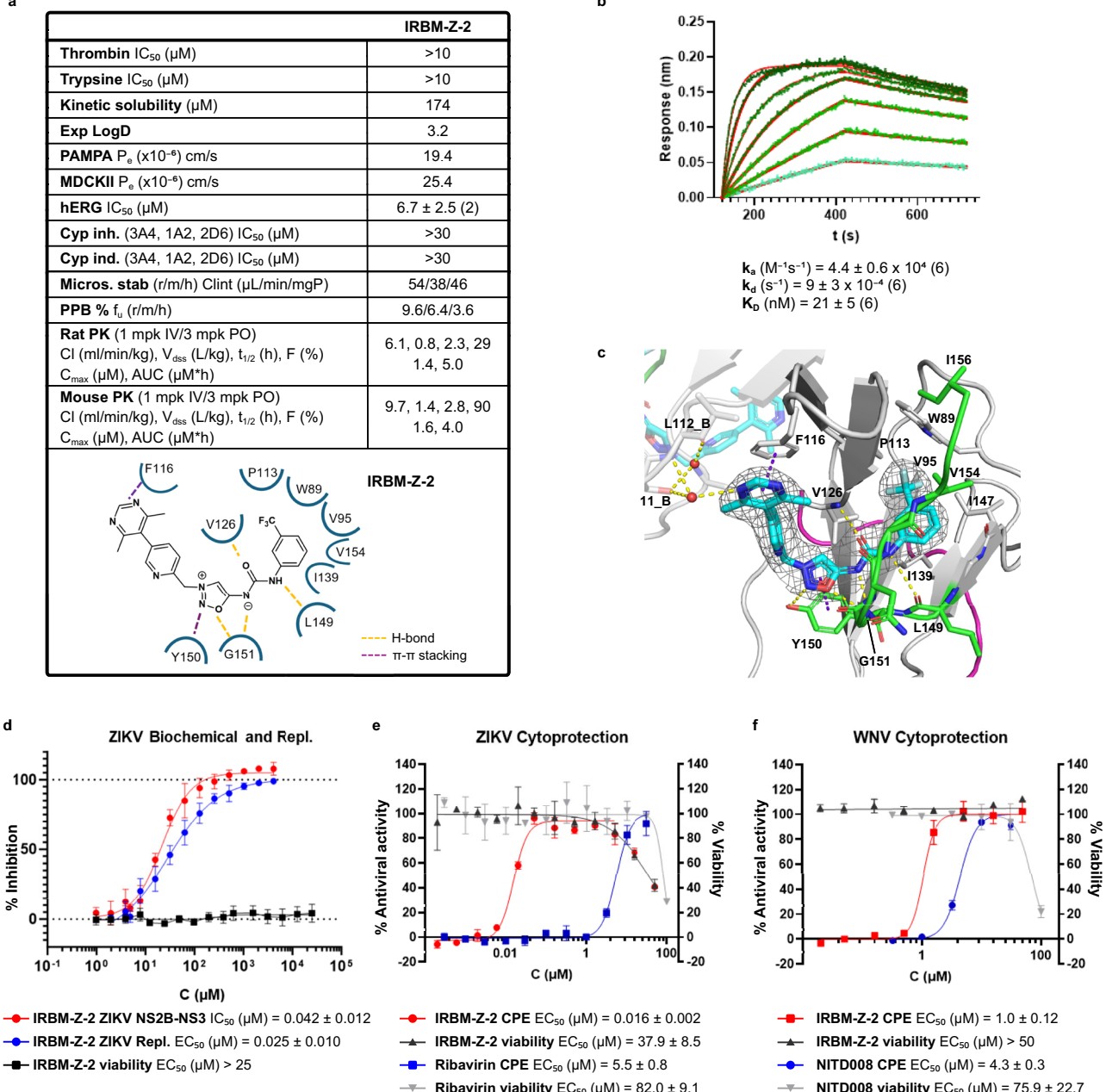

**Fig. 3 | IRBM-Z-2 structure, binding mode and biological characterization.**
**a** Chemical structure of IRBM-Z-2 along with biological data and in vitro and in vivo ADME profiles. Two-dimensional (2D) interaction patterns of IRBM-Z-2 and NS2B-NS3 protease depicted by dashed lines. **b** Representative BLI (biolayer interferometry) sensorgrams (blue curves) with 1:1 fitting (red curves) of the IRBM-Z-2 and ZIKV NS2B-NS3 protease interaction. Binding parameters represent the mean ± SD of six independent experiments. **c** Close-up view of the allosteric binding mode of IRBM-Z-2 (cyan sticks) co-crystallized in complex with the ZIKV NS2B-NS3 protease (PDB ID 9IBY). The ligand is depicted in cyan sticks; the H-bonds are depicted as dashed yellow lines and the π-stacking interactions as dashed purple lines. The second monomer is depicted as transparent sticks and cartoons. Electron density difference map $2F_O$-$F_C$, at +1σ carved at 1.5 Å from the ligand is shown as grey isomesh. **d** Dose−response curves and $IC_{50}$ and $EC_{50}$ values of IRBM-Z-1 in the NS2B-NS3 protease enzymatic assay (red curve) and ZIKV replicon assay (blue curve), and Vero cell proliferation assay (black curve). All data represent the mean ± SD from at least three independent experiments. **e** Dose-response curves showing antiviral activity and cytotoxicity of IRBM-Z-2 and Ribavirin against ZIKV (PRVABC59 strain) infected BHK-21 cells. Antiviral activity is measured as a % reduction of cytopathic effect (CPE); cytotoxicity was assessed in parallel. **f** Dose-response curves showing antiviral activity and cytotoxicity of IRBM-Z-2 and NITD008 against WNV infected Vero cells. The analysis is presented as % luminescence signal (indicative of viral replication). Cytotoxicity was evaluated using the XTT assay. All graphs represent the mean ± SD. from three biological replicates (*n* = 3).

Furthermore, these agents could lead to the development of combination therapies, a highly effective and safe strategy used against other viruses such as Hepatitis C virus (HCV)[24]. To this end, we developed a replicon-based assay for ZIKV to support an HTS campaign aimed at identifying small molecules with a novel mechanism of action. This phenotypic approach led to the rapid identification of a new cell-permeable class of small molecules represented by our hit compound, IRBM-Z-1, that shows potent antiviral activity in cell-based assays at low micromolar concentrations.

Sequencing of IRBM-Z-1-resistant ZIKV replicons revealed a single mutation, I156T in the NS3 protease, that abolished viral replication, which strongly indicated that NS3's enzymatic activity is the primary

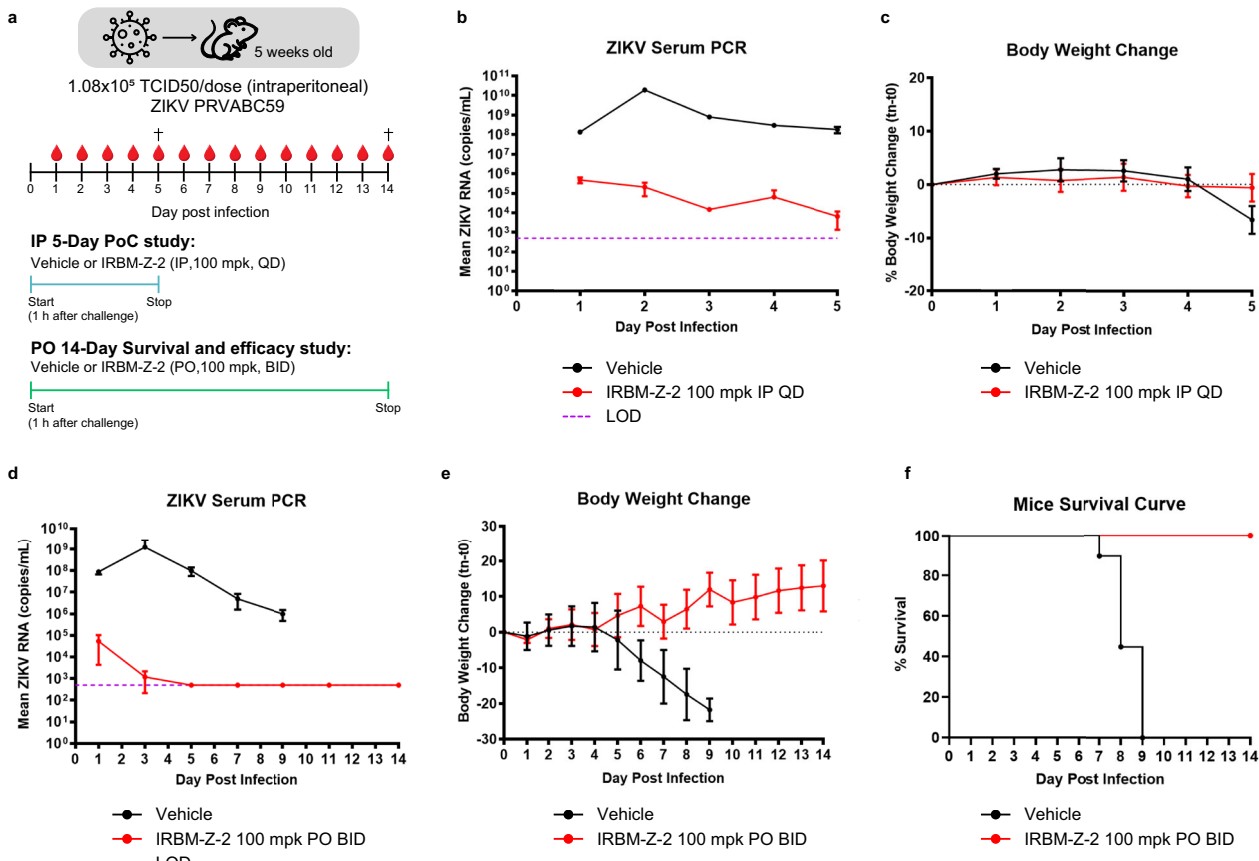

**Fig. 4 | In vivo efficacy of IRBM-Z-2 against ZIKV infection in AG129 mice.**
**a** Schematic diagram of the IP 5-day PoC and PO 14-day survival efficacy studies. In both, AG129 mice were infected intraperitoneally (IP) with ZIKV viral stock. Each treatment arm was divided for blood collection into three subgroups on alternating days (IRBM-Z-2: $n = 3$ per group for IP and PO; vehicle: $n = 2$ per group for IP; $n = 3$ per group for PO). **b, c** IP 5-day PoC study: infected mice were treated 1 h post-infection with IRBM-Z-2 IP (100 mg/kg, QD, $n = 9$) or vehicle ($n = 6$). Efficacy was assessed daily for 5 days based on viral load (**b**) and body weight change (**c**). **d**–**f** PO

14-Day Survival and efficacy study: infected mice were treated orally with IRBM-Z-2 (100 mg/kg BID, 200 mg/kg/day) or vehicle starting 1 h post-infection for 14 consecutive days ($n = 9$ per group). Efficacy was assessed by viral load (**d**), body weight change (**e**), and survival (**f**). Serum viral load was quantified by RT-qPCR and expressed as $\log_{10}$ viral RNA copies per mL. The dotted line indicates the lower limit of detection (~500 copies/mL). Data points represent the mean ± s.d. values for each group. †Indicates day of euthanasia. Schematic diagram in (**a**) was adapted from refs. [20] and [21].

target. Remarkably, I156T was the only mutation identified in the resistance generation experiments. This finding suggests that the mutation does not significantly impair replication and is the most readily selected among potential variants under our experimental conditions. Notably, the I156T variant has not been observed in any of the natural ZIKV isolates, indicating minimal risk of pre-existing resistance. Mechanistic and kinetic studies confirmed that the identified chemical series functions as non-competitive allosteric inhibitors of the NS3 protease. The NS2B-NS3 protease complex is essential for ZIKV replication, as it cleaves the viral polyprotein into the individual functional proteins required for viral maturation. The NS2B cofactor wraps around the NS3 protease domain, stabilizing its active conformation and enabling proteolytic activity. Given its central role in the viral life cycle, the NS3 protease is a well-validated and attractive target for antiviral drug development. However, most competitive inhibitors reported to date have either a peptidic structure, with strong basic residues, which results in poor cell permeability and limited in vivo antiviral efficacy, or are small molecules with low inhibitory potency[17,25–29]. The favourable molecular properties of our compounds overcome these limitations and opens the road to a new series of potent antiviral drugs. The crystal structure of the NS2B-NS3 protease in complex with $R$-( + )-IRBM-Z-1, revealed that this inhibitor class is characterized by a unique $N$-acylsydnone imine ring that forms an

extensive network of H-bonds with the protease backbone, preventing proper folding of the NS2B domain and locking the enzyme in a catalytically inactive conformation. This $N$-acylsydnone imine moiety is not commonly found in drug-like molecules[30], and its structural novelty and unique binding properties suggests broader applicability in the design of inhibitors targeting challenging protein pockets by establishing additional H-bond interactions. Through iterative optimization, we identified IRBM-Z-2, a nanomolar inhibitor of ZIKV replication, with no associated cytotoxicity. Importantly, no efflux liabilities were observed. In two murine models of ZIKV infection, IRBM-Z-2 showed strong prophylactic efficacy via both intraperitoneal and oral administration, resulting in viral loads more than 4 $\log_{10}$ lower than controls. This robust in vivo efficacy positions IRBM-Z-2 among the more effective pre-clinical or clinical antiviral candidates currently under investigation[25,26]. Notably, IRBM-Z-2 also inhibited the NS3 proteases of both DENV2 and WNV at low micromolar to nanomolar concentrations, even though the potential efficacy against DENV needs to be evaluated further. The development of a broad-spectrum pan-orthoflavivirus inhibitor remains an attractive yet challenging goal. Our findings support the potential of this compound as a promising starting point to design potent inhibitors targeting the same mechanism of action in other orthoflaviviruses, including DENV2 and WNV. We are currently conducting follow-up studies to identify more

advanced compounds, whose antiviral efficacy will be evaluated against additional ZIKV strains as well as other orthoflaviviruses, both in cell-based systems and relevant animal models. The results of these ongoing investigations will be disclosed in future publications.

In conclusion, we have identified a novel series of potent non-competitive allosteric inhibitors of the ZIKV NS2B-NS3 protease. These compounds exhibit potent antiviral activity in both in vitro and in vivo models, with excellent drug-like properties and safety profiles. This study establishes a new framework for the development of next-generation antiviral drugs against ZIKV and other related orthoflaviviruses.

## Methods
### Ethics statement
This study was performed under an Institutional Animal Care and Use Committee (IACUC)-approved protocol. Housing and handling of the animals were performed in accordance with the standards of the American Association for Accreditation of Laboratory Animal Care (AAALAC) International's reference resource: the 8th edition of the Guide for the Care and Use of Laboratory animals, Animal Welfare Act as amended, and the 2015 reprint of the Public Health Service (PHS) Policy on Human Care and Use of Laboratory Animals. Handling of samples and animals occurred in compliance with the Biosafety in Microbiological and Biomedical Laboratories (BMBL), 5th edition (Centers of Disease Control).

### Compound collection
CNCCS (www.cnccs.it) is a public-private consortium whose objective is the development of a compound collection that facilitates high-throughput screening, enabling researchers to rapidly test thousands of compounds against biological targets and identify potential drug candidates, with a focus on rare, neglected, and poverty-related diseases (https://www.cnccs.it/en/collezione-nazionale-dei-composti-chimici/raccolta-composti/). The collection contains approximately 120,000 compounds, including FDA- and EMA-approved drugs. It is designed for broad applicability, with no bias toward specific targets or diseases, and optimized for favourable physicochemical properties such as calculated logD, sp³ character, hydrogen bond donors/acceptors, and polar surface area.

### Cells
Vero cells (African green monkey kidney fibroblasts; Sigma-Aldrich, Cat. #84113001 ECACC) were cultured in DMEM (Dulbecco's Modified Eagle Medium) with high glucose and pyruvate (Gibco, Cat. #41966), supplemented with 10% FBS (fetal bovine serum) (Gibco, Cat. #10270) and 1% penicillin-streptomycin (10 mg/mL; Gibco, Cat. #14140).

### Generation of ZIKV, WNV, and DENV2 replicons
For the ZIKV subgenomic replicon construction, we used the sequence of a clone of the Natal RGN isolate of Asian lineage (GenBank: KU527068.1), It was modified by deleting 2193 nucleotides encoding structural proteins, except for a fusion comprising the 31 N-terminal amino acids of the capsid and the 32 C-terminal amino acids of the envelope protein. These sequences were retained to preserve the correct processing and translocation of NS1 and of the non-structural polyprotein across the endoplasmic reticulum membrane. An EMCV IRES sequence was inserted downstream of the polyprotein stop codon, driving expression of a NanoLuc-UBI-NEO fusion protein for dual reporter and selection purposes. The ZIKV annotated replicon sequence is reported in Supplementary Fig. 4.

The WNV replicon was generated from the U.S.A. isolate (GenBank: AY646354.1) with similar modifications, retaining 31 N-terminal residues of the capsid and 30 C-terminal residues of the envelope. The IRES drove expression of a NanoLuc-UBI-NEO fusion.

The DENV2 replicon was based on the type 2 New Guinea C isolate (GenBank: AF038403.1). In this replicon, the coding sequences of structural genes were deleted except for 36 amino acids at the N-terminus of the capsid protein that were fused to a NanoLuc-UBI-NEO fusion protein. An EMCV IRES was inserted after the stop codon of the NEO gene driving a 34 amino acid portion of the C-terminus of the E protein followed by the rest of the DENV2 genome.

The replicon constructs were transcribed from T7 promoters and transfected into Vero cells. Stable cell lines were selected in 0.7 mg/mL G-418 (Sigma-Aldrich, Cat. #4727894001) until clonal populations could be expanded.

### Replicon inhibition and cell viability assays
Assays were performed as previously described[29]. Replicon-expressing Vero cells were seeded (5000–6000 cells/well) in 384-well black plates (Greiner, Cat. #781086) containing compounds (10 μM for HTS; serial dilutions for dose-response) dispensed via an acoustic liquid handler (ATS-100 EDC or Echo650, Beckman). Positive controls included mycophenolic acid[31] (50 μM) and NITD008[32–35] (50 μM) (Supplementary Fig. 1). Gambogic acid (32 μM; Sigma-Aldrich, Cat. #G8171) was used as a cytotoxicity control. Negative control wells received 0.5% DMSO.

After 72 h incubation at 37 °C with 5% $CO_2$, the NanoLuc signal was revealed using the Nano-Glo reagent (Promega, Cat. #N1150), and cell viability was assessed using CellTiter-Glo (Promega, Cat. #G7573). Luminescence was measured using an Envision reader (PerkinElmer) 10 min after Nano-Glo and 30 min after CellTiter-Glo. $EC_{50}$ values were calculated by four-parameter logistic regression using Dotmatics or GraphPad Prism. The dose-response curve and $EC_{50}$ for compound IRBM-Z-1 are shown in Fig. 1c and Table 1, and for compound IRBM-Z-2 in Fig. 3d and Table 1.

### Selection of resistant mutants
Replicon-containing cells were cultured in the presence of 5 x $EC_{50}$ concentrations of candidate compounds. For IRBM-Z-1, cells were exposed to 30 μM until resistant clones emerged. Total RNA was extracted (Qiagen), and full-length replicon sequencing was performed by NGS (or Sanger for specific region).

### Compound synthesis and characterization
Synthetic routes and chemical characterization data for IRBM-Z-1, R-(+)-IRBM-Z-1, S-(-)-IRBM-Z-1 and IRBM-Z-2 are described in the Supplementary Methods. Synthesis of the ZIKV protease inhibitor CN-716 followed a literature-reported route[9]. RdRp inhibitor NITD008 was purchased from Tocris Biosciences (Cat. #6045), and mycophenolic acid from Sigma-Aldrich (Cat. #M5255). All compounds were dissolved in 100% DMSO to a 10 mM stock concentration for in vitro use.

### ZIKV, DENV2 and WNV NS2B-NS3 protease inhibition assays
The ZIKV NS2B-G4SG4-NS3 protease was expressed and purified as previously described[29]. The protease (1.25 nM) was incubated with serial dilutions of compounds (0.097–100 μM) in assay buffer (50 mM Tris-HCl, pH 8.5, 1% glycerol, 1 mM CHAPS, 1% DMSO) for 10 min at 25 °C in 384-well black plates (Greiner, Cat. #781900). The substrate Bz-Nle-KRR-AMC (10 μM; Bachem, Cat. #4055312) was added and incubated for 30 min. Product fluorescence (Ex 360 nm/Em 465 nm) was read on a SPARK TM10 (Tecan).

The DENV2 NS2B-G4SG4-NS3 protease was expressed and purified as previously described[29] and tested at 10 nM using the same protocol as above, with 15 μM Bz-Nle-KRR-AMC as the substrate.

The WNV NS2B-NS3 protease (R&D Systems, Cat. #2907-SE-020) was used per the manufacturer's instructions and tested at 2 nM using the same protocol as above with minor modifications. The reaction buffer included 50 mM Tris-HCl (pH 9.0), 30% glycerol and 1% DMSO.

The substrate was 10 μM pERTKR-AMC (R&D, Cat. #ES013). Data analysis was as described for ZIKV.

Results were analysed using Prism (GraphPad, San Diego, CA) and Vortex software (Dotmatics, Bioshops Stortford, UK). $IC_{50}$ values were calculated by four-parameter logistic regression. For mechanism-of-action studies, protease inhibition was assessed at multiple substrate concentrations (1.875-120 μM), and $IC_{50}$ values were plotted against [S]/KM using GraphPad Prism.

### Human thrombin and trypsin inhibition assay

Thrombin and trypsin inhibitory activity was assessed by Eurofins Panlabs Discovery Services Ltd (Taiwan), https://emea.eurofinsdiscovery.com/catalog/thrombin-human-chymotrypsin-serine-peptidase-enzymatic-leadhunter-assay-tw/165000 and https://emea.eurofinsdiscovery.com/catalog/trypsin-non-selective-human-chymotrypsin-serine-peptidase-enzymatic-leadhunter-assay-tw/165100, respectively. Human plasma thrombin (Accession number: NM_000506) and human pancreatic trypsin (Accession number: NM_002769) were used for the selectivity assays. Test compounds or vehicle controls were pre-incubated with thrombin (8.25 mU ml$^{-1}$) or trypsin (0.016 nM) in modified HEPES buffer (pH 7.5) for 15 min at 25 °C. Reactions were initiated by addition of the fluorogenic substrate Z-Gly-Pro-Arg-AMC (20 μM) and allowed to proceed for 60 min at 25 °C. Formation of AMC was quantified spectrofluorometrically (excitation 360 nm, emission 465 nm). Compounds were tested at 10 μM in triplicate.

### ZIKV NS3 binding kinetics by BLI

Binding of IRBM-Z-1 and IRBM-Z-2 to ZIKV NS2B-NS3 was measured using an OctetRed96e instrument (Sartorius) at 24 °C. The protease was biotinylated (Thermo, Cat. #21338) using a 3-fold molar excess of biotin at 4 °C for 18 h, followed by desalting (Zeba Spin Columns, Thermo, Cat. #89883). Biotinylated protein (30 μg/mL in HBS-P + , 0.1% BSA) was immobilized on streptavidin biosensors for 600 s (ForteBio, Cat. #18-5019), followed by biocytin saturation (10 μg/mL, 1 min). A column of eight biosensors was left blank to serve as reference surface. Compound binding was assessed by 8-point titration (IRBM-Z-1: 10-0.156 μM; IRBM-Z-2: 1–0.015 μM) in HBS-P$^+$, 0.1% BSA, 1% DMSO. Association/dissociation times were 300 s each. Data were collected at 5 Hz, double-referenced and fit to a 1:1 Langmuir model interaction model using the global data analysis option available within HT 11.1 software (Sartorius).

### Crystallization of ZIKV NS2B-NS3 complexed with IRBM-Z-1 or IRBM-Z-2

The ZIKV NS2B-NS3 protease was expressed in *E. coli* as a fusion construct comprising residues 45–96 of NS2B and residues 1–177 of NS3, linked via a flexible GGGGSGGGG linker. An N-terminal hexahistidine tag with a thrombin cleavage site was included to facilitate purification. Expression was performed in LB medium at 18 °C for 18 h at a 10 L scale. Cells were lysed by sonication in lysis buffer (30 mM Tris-HCl, pH 8.0, 500 mM NaCl, 10 mM imidazole, 5% glycerol, 2 mM β-mercaptoethanol). The lysate was clarified by centrifugation and subjected to immobilized metal affinity chromatography (IMAC), followed by gradient elution to 500 mM imidazole.

Protein-containing fractions were pooled, concentrated and further purified via size-exclusion chromatography (SEC) on a HiLoad 26/600 Superdex 75 column in SEC buffer (20 mM HEPES, pH 7.5, 150 mM NaCl, 5% glycerol, 2 mM DTT). The pooled protein was digested with thrombin overnight at 4 °C while dialyzing against 30 mM Tris-HCl (pH 8.0), 50 mM NaCl and 2 mM DTT. Uncleaved protein was removed by subtractive IMAC. The flow-through was loaded onto a ResourceQ anion exchange column equilibrated in buffer A (30 mM Tris-HCl, pH 8.0, 50 mM NaCl, 1 mM DTT) and eluted using a linear gradient to

buffer B (same as A, but with 1 M NaCl). Final polishing was performed using SEC in the same buffer. Purity (>95%) was confirmed by SDS-PAGE with Coomassie staining.

The purified NS2B-NS3 protease was concentrated to 20–24 mg/mL and incubated with either *R*-( + )-IRBM-Z-1 or IRBM-Z-2 for 1–3 h prior to crystallization. Co-crystals with *R*-( + )-IRBM-Z-1 were grown using sitting-drop vapor diffusion (1:1 drop ratio) in a condition containing isopropanol, Li$_2$SO$_4$ and sodium phosphate–citrate buffer (pH 4.25). Crystals were optimized via streak seeding. Co-crystals with IRBM-Z-2 were obtained under similar conditions using PEG 8000, glycerol, magnesium acetate and sodium cacodylate (pH 6.25), with streak seeding for optimisation.

Crystals were cryoprotected in 20% glycerol, flash-frozen in liquid nitrogen, and data were collected at 100 K. X-ray diffraction data for the *R*-( + )-IRBM-Z-1 complex (9TPG) were collected at the Swiss Light Source (SLS, Villigen, Switzerland), while data for the IRBM-Z-2 complex (9IBY) were collected at the European Synchrotron Radiation Facility (ESRF, Grenoble, France). Data were processed using autoP-ROC, XDS, and AIMLESS. Data collection and processing statistics are reported in Supplementary Table 3.

Phase information was obtained by molecular replacement using a previously determined structure of the ZIKV NS2B-NS3 protease as the search model. Molecular replacement and subsequent structure solution were performed using the CCP4 suite. Iterative model building was carried out in COOT, and refinement was performed using REFMAC5 within the CCP4 package. Translation/Libration/Screw (TLS) refinement was applied, resulting in improved R-factors and enhanced electron density quality. Local non-crystallographic symmetry (NCS) restraints were generated automatically using the "ncsr local" keyword in REFMAC5. Ligand topology and parameter files were generated using CORINA, and model restraints were incorporated during refinement. Both co-crystal structures were refined against diffraction data processed with STARANISO to account for anisotropy, where applicable.

Refinement statistics and model quality metrics are summarized in Supplementary Table 3.

### ZIKV infection assay in Vero cells (IRBM-Z-1)

The antiviral activity of IRBM-Z-1 was evaluated in Vero cells by measuring inhibition of ZIKV-induced cytopathic effects (CPE). Assays were performed by RetroVirox Inc. (San Diego, USA) using the PLCal_ZV strain (human\2013\Thailand). Vero cells (*Cercopithecus aethiops* kidney epithelial, 84113001, Sigma) were maintained in DMEM supplemented with 5% FBS (DMEM5) and seeded at 10,000 cells per well in 96-well plates.

After 24 h incubation at 37 °C, test compounds were diluted 5-fold in MEM supplemented with 1% bovine serum albumin (MEM1-BSA) and pre-incubated with cells for 30 min at 37 °C.

ZIKV was added to each well and incubated for 3 h, followed by washing with MEM containing 2% FBS (MEM2) to remove unbound virus. Fresh test compound dilutions were then added into MEM2 supplemented with 0.05% DMSO. Plates were incubated for 6 days at 37 °C and 5% CO$_2$. Cell viability was determined using a neutral red uptake assay: cells were stained with 0.017% neutral red for 3 h, washed, and lysosomal dye was extracted using 50% ethanol with 1% acetic acid. Absorbance at 540 nm was measured to quantify viable cells.

Test compounds were evaluated in duplicate using serial 5-fold dilutions. Controls included mock-infected cells, virus-infected cells without compound (vehicle control), wells with test compound only (cytotoxicity control), and mycophenolic acid (MPA) as positive control. Inhibition of virus-induced CPE was calculated by normalizing absorbance to mock-infected and virus-only controls. $EC_{50}$ values were derived from four-parameter nonlinear regression analysis.

### ZIKV infection assay in BHK-21 cells (IRBM-Z-2)

The antiviral activity of IRBM-Z-2 against ZIKV (PRVABC59 strain) was evaluated by measuring inhibition of ZIKV-induced cytopathic effects (CPE) in BHK-21 cells (CCL-10, ATCC). Assays were performed by ImQuest Biosciences Inc. (Frederick, USA). The ZIKV PRVABC59 strain was obtained from ATCC (VR-1843) and propagated in LLC-MK2 cells to generate virus stocks. BHK-21 cells were maintained in DMEM supplemented with FBS, 2 mM L-glutamine, 100 U/mL penicillin and 10 μg/mL streptomycin. BHK-21 cells ($3 \times 10^3$ cells per well) were seeded in 96-well flat-bottom tissue culture plates and allowed to adhere overnight at 37 °C in a humidified 5% $CO_2$ incubator. The following day, diluted test compounds and virus (at a concentration predetermined to induce 85–95% cell death by day 6 post-infection) were added to the wells. Plates were incubated for 6 days under standard culture conditions. Cell viability was quantified using XTT tetrazolium dye (Roche), with absorbance measured at 450 nm (reference 650 nm) using a plate reader (SoftMax Pro 4.6). Percent CPE inhibition and cell viability were calculated relative to virus and cell control wells. $EC_{50}$ and $TC_{50}$ values were determined by non-linear four-parameter dose-response curve fitting.

### ZIKV infection assay in HuH-7 cells (IRBM-Z-2)

Antiviral assays with IRBM-Z-2 against ZIKV (Padova and H/PF/2013 strains) were performed by Artemis Bioservices B.V. (Delft, The Netherlands). HuH-7 cells (CL-0120, ElabScience) were seeded in 96-well plates at $1.2 \times 10^4$ cells per well and incubated for 16 h prior to infection. IRBM-Z-2 was diluted to 25 μM from a 50 mM stock solution, followed by preparation of an eight-point, four-fold serial dilution. Each dilution was pre-mixed with 1000 $TCID_{50}$ virus per well and incubated for 10–15 min at room temperature. The virus–compound mixtures (100 μL per well) were then added to the cells and incubated for 1.5 h at 37 °C with 5% $CO_2$. After adsorption, wells were washed twice with pre-warmed PBS and replenished with fresh medium containing the corresponding compound concentrations to maintain drug pressure. Plates were incubated for 48 h at 37 °C with 5% $CO_2$. Cells were fixed with 5% paraformaldehyde and permeabilized using a 1:1 mixture of ice-cold 0.1% Triton X-100 and 70% ethanol. Infected cells were detected using a ZIKV-specific primary antibody and an Alexa Fluor 488-conjugated secondary antibody (mouse anti-flavivirus envelope protein IgG2a monoclonal antibody (4G2)(AbFLAVENV-4G2-200, dilution factor 1:500) and imaged with a Cytation™ V1 Cell Imaging Multi-Mode Reader (BioTek) using Gen5 software to quantify virus-positive cells. Cytotoxicity was assessed in parallel by MTT assay (3-[4,5-dimethylthiazol-2-yl]-2,5-diphenyltetrazolium bromide), which measures cellular metabolic activity.

### WNV infection assay in Vero76 cells (IRBM-Z-2)

The antiviral efficacy of IRBM-Z-2 against WNV was assessed by the Institute for Antiviral Research (Utah State University) using a neutral red uptake assay in Vero76 cells (CRL-1587, ATCC). The WNV isolate used (strain WN02, designated Kern 515) was originally recovered from a mosquito collected in Kern County and was obtained from the University of Texas Medical Branch Arbovirus Reference Collection (TVP 10799, BBRC lot no. WNVKERN515-01). Test compounds were serially diluted in Minimum Essential Medium (MEM) supplemented with 2% FBS and 50 μg/mL gentamicin, using eight half-log dilutions. Diluted compounds were added to 96-well plates containing 80–100% confluent Vero76 cells. For each concentration, three replicate wells were infected with WNV, while two wells were uninfected to assess compound cytotoxicity. Virus-only and cell-only controls (six wells each) were included on each plate. The WNV inoculum was calibrated to achieve > 80% CPE at six days post-infection. Following a 6-day incubation at 37 ± 2 °C with 5% $CO_2$, plates were stained with neutral red for 2 h. After washing with PBS, intracellular dye was extracted using 50:50 Sorensen citrate buffer and ethanol. Optical density was measured at 540 nm. Absorbance values were normalized to uninfected cell controls and virus-only controls. $EC_{50}$ and $CC_{50}$ values were calculated by regression analysis, and the selectivity index (SI) was computed as the ratio of $CC_{50}$ to $EC_{50}$. INFERGEN (interferon alfacon-1) was used as a positive control. In a complementary assay, virus-induced cytotoxicity and compound toxicity were quantified using the XTT tetrazolium dye method. Cell viability was expressed relative to untreated controls, and $EC_{50}$ values were determined by non-linear regression.

### In vivo pharmacokinetic studies

All animal studies were conducted in full compliance with the EU Directive 63/2010 (On the Protection of Animals used for Scientific Purposes) and its Italian transposition (Italian Decree no. 26/2014) as well as with all applicable Italian legislation and guidelines. In particular, this project was submitted for comments and approval to the internal IRBM Ethics Committee and then submitted to the Italian Ministry of Health for government authorization. The animal facility is authorized by the Italian Ministry of Health and by a local veterinary authority to house and use laboratory rodents for scientific purposes (Authorization 22/2023 UT). Pharmacokinetic (PK) studies were performed in male Sprague–Dawley rats and male C57BL/6 mice. In rats, test compounds were administered intravenously (i.v.) at 1 mg/kg, formulated in 20% (w/v) 2-hydroxypropyl-β-cyclodextrin in 50 mM citrate buffer (pH 5.5) at a concentration of 1 mg/mL. Plasma samples were collected by serial microsampling at 0.083, 0.25, 0.5, 1, 2, 4, 8 and 24 h post-dose. For oral administration, compounds were dosed by gavage at 3 mg/kg using the same vehicle, and plasma was collected at 0.25, 0.5, 1, 2, 4, 8 and 24 h post-dose.

In mice, IRBM-Z-2 was administered i.v. at 1 mg/kg (0.2 mg/mL in vehicle) and orally at 3 mg/kg using the same vehicle as in rats. Blood samples were collected at the same time points as described above via serial microsampling.

### ZIKV infection models in mice

Mouse efficacy studies were conducted under optimized and validated conditions at BIOQUAL, Inc. (Rockville, MD, USA) in accordance with the *Guide for the Care and Use of Laboratory Animals*, the U.S. Animal Welfare Act regulations, and Office of Laboratory Animal Welfare (OLAW) guidelines. The study was performed under an Institutional Animal Care and Use Committee (IACUC)-approved protocol (23-034 P). The experimental design incorporated an appropriate number of animals per group (n ≥ 9 treated mice), consistent with previously published models of ZIKV infection[20,21]. Mouse studies were conducted in AG129 mice (129/Sv strain deficient in both IFNα/β and IFNγ receptors) which were bred and maintained under specific pathogen-free conditions at the BIOQUAL facility.

During the acclimation period and immunization phase, the animals were housed up to 5 mice per cage within their respective study groups. Environmental controls for the animal room were set to maintain ~20 °C to 26 °C, a relative humidity of 30–70%, and a 12 h light/12 h dark cycle according to BIOQUAL standard procedures. All animals were provided free access to water by a water bottle, and rodent diet (Lab Diet 5021) was provided with ad libitum, unless otherwise specified according to BIOQUAL standard procedures. Samples of the water were analysed by BIOQUAL for specified microbiological content and for chlorine level, pH, hardness, and heavy metals. The food was routinely analysed by the manufacturer for nutritional components and environmental contaminants. BIOQUAL provides environmental enrichment in accordance with Office of Laboratory Animal Welfare (OLAW) requirements, U.S. Animal Welfare Act regulations, and in compliance with the Guide for the Care and Use of Laboratory Animals. BIOQUAL has developed and implemented a science-based environmental and behavioural enrichment program plan.

IP 5-Day PoC study was conducted as follows. Fifteen male AG129 mice (5 weeks old) were randomized into six groups. Groups 1–3 (n = 2 per group) received vehicle control, while Groups 4−6 (n = 3 per group) received IRBM-Z-2. On Study Day (SD) 0, all animals were challenged intraperitoneally (IP) with $1.08 \times 10^5$ $TCID_{50}$ of ZIKV PRVABC59 in 100 μL volume. One hour post-challenge, mice were administered either vehicle (20% w/v 2-hydroxypropyl-β-cyclodextrin in 50 mM citrate buffer, pH 5.5) or IRBM-Z-2 (100 mg/kg in the same vehicle, 10 mg/mL) via IP injection. Treatment was continued once daily from SD1 to SD4. Study endpoints included daily body weight measurements, clinical scoring, and serum viral load analysis via RT-qPCR (see study design in Supplementary Table 5).

PO 14-Day Survival and efficacy study was conducted as follows. Eighteen male AG129 mice (5 weeks old) were randomized into six groups. Groups 1-3 (n = 3 per group) received vehicle control, while Groups 4−6 (n = 3 per group) received IRBM-Z-2. All animals were challenged on SD0 via IP injection with $1.08 \times 10^5$ $TCID_{50}$ of ZIKV PRVABC59 in 100 μL. At 1 h post-challenge, mice in Groups 4−6 were administered 100 mg/kg IRBM-Z-2 via oral gavage (PO), with an additional dose at 13 h post-challenge. From Day 1 through Day 13 post-infection (d.p.i.), treatment was administered twice daily (BID). Body weights were measured daily and during blood collection, sedation, or when clinically indicated. Blood samples were collected on d.p.i. 1 and 7 (Group 4), d.p.i. 3 and 9 (Group 5), and d.p.i. 5 and 11 (Group 6). Terminal samples were collected either at euthanasia or on d.p.i. 14 (see study design in Supplementary Table 6). For specimen processing and RNA extraction, whole blood was incubated at room temperature for ≥ 30 min to allow clot formation. Samples were centrifuged for 5 min, and serum was collected and stored at -80 °C until analysis. Viral RNA was extracted from 200 μL of serum using the QIAamp Viral RNA Mini Kit (QIAGEN) per the manufacturer's instructions. Quantification of ZIKV RNA was performed using a one-step reverse transcription quantitative PCR assay (forward primer (5′GGAAAAAAGAGGC TATGGAAATAATAAAG reverse primer (5′CTCCTTCCTAGCATTGAT TATTCTCA and probe (5′AGTTCAAGAAAGATCTGGCTG) on a 7500 Real-Time PCR System (Applied Biosystems). Viral load and survival data were analysed using GraphPad Prism (v9.0 or higher). Group comparisons were performed using two-way ANOVA with Tukey's post hoc test or log-rank (Mantel-Cox) tests, as appropriate. Statistical significance was defined as $P < 0.05$.

## Human dose determination
Human pharmacokinetic predictions were performed using in vitro-in vivo extrapolation (IVIVE) methods. Intrinsic clearance was measured in plated human cryopreserved hepatocytes (10 donors, mixed gender) purchased from BioIVT (Germany) (Supplementary Table 4), and hepatic clearance was estimated using the well-stirred liver model. Volume of distribution at steady state (Vss) was estimated from preclinical species data and corrected for human plasma protein binding. Oral bioavailability (F) was calculated as the product of hepatic availability (Fh) and absorption multiplied by gut metabolism (Fa × Fg), where Fh was derived from IVIVE clearance and Fa × Fg was based on preclinical PK data.

## Statistical analysis and data representation
Data from the biochemical and cell-based assays were analysed using Prism (GraphPad, San Diego, CA, Version 10.6.1), Dotmatics (Bioshops Stortford, UK, Version 5.0) and Vortex software (as apert of Dotmatics, Bioshops Stortford, UK, Vortex 6.1.2.1372-s). $IC_{50}$ values were calculated by four-parameter logistic regression. For mechanism-of-action studies, protease inhibition was assessed at multiple substrate concentrations: $IC_{50}$ determination and correlation between these values and [S]/KM were analysed using GraphPad Prism. Data from the BLI kinetics studies were double-referenced and fit to a 1:1 Langmuir model

using HT 11.1 software (Sartorius). The program GraphPad Prism was also used to create the scientific graphs of this work.

## Data availability
All data are available within the main text or the supplementary materials. Atomic coordinates and structure factors for the co-crystal structures have been deposited in the PDB under accession codes 9TPG and 9IBY. The synthesis and chemical characterization of compounds IRBM-Z-1, R-(+)-IRBM-Z-1, S-(-)-IRBM-Z-1 and IRBM-Z-2, including ¹H, ¹⁹F and ¹³C NMR data, is described in the Supplementary Information. All source data are provided with this paper as a Source data file. Source data are provided with this paper.

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

## Acknowledgements

C. Iaccarino, V. Nardi, L. Mangiardi and L. Lazzaro for compound purification and QC analysis. M. R. Battista for microsome stability assays, V. Iannizzotto for the CYP induction and Hepatopac assays, I. Carlì, M. Luche, M. A. Sweeney and M. Sadowski for scientific writing corrections and figure preparation assistance. The crystal structure of *R*-(+)-IRBM-Z-1 (PDB ID 9TPG) and IRBM-Z-2 (PDB ID 9IBY) with ZIKV NS2B-NS3 protease were determined at Proteros Biostructures GmbH (Martinsried, Germany), and we appreciate the efforts and support of Stephan Krapp, Regina Freier, and Sina Kordes at Proteros Biostructures GmbH. IRBM S.p.A. acknowledges financial support from the Italian National Compound and Screening Collection (CNCCS consortium) through the "Identification of new molecules with therapeutic potential for the treatment of the ZIKV and other emerging viruses" project (funded by Regione Lazio 2018–2021).

## Author contributions

J.M.O., G.P., E.T., C.A. and A.M. wrote the original draft. Compound synthesis and target design were carried out by A.Q., F.F., A.C., G.I., L.B., E.T., R.Di F. and J.M.O.; CADD support by J.A. and S.V.; biological studies by R.G., S.C., M.B., A.M., C.A., E.B., C.P., N.A. and G.P.; DMPK studies by M.V., M.N., M.V.O., G.P., F.C., L.O. Project overview was performed by A.B., G.P., J.M.O., R.Di F., L.T., V.P., C.M., and C.T. All authors have given approval to the final version of the manuscript.

## Competing interests

IRBM-Z-1, R-(+)-IRBM-Z-1, S-(-)-IRBM-Z-1 and IRBM-Z-2 are covered by patent WO2023/227734 (IRBM S.P.A. and C.N.C.C.S., PCT/EP2023/064095) with the following co-authors listed as inventors: C.A., A.B., R. Di F., C.M., J.M.O., G.P., A.Q., E.T., L.B., J.A., A.C. and F.F. An additional patent application related to WO2023/227734 is WO2025/114399 (IRBM S.p.A. and C.N.C.C.S., PCT/EP2024/083832) and another one is pending. The other authors declare no competing interests.
