## [Transparent Peer Review file · Nature Communications]

An allosteric inhibitor of the Zika virus NS2B-NS3 protease with oral efficacy in mouse models

Corresponding Author: Dr Jesus Ontoria

Version 1:

Reviewer comments:

Reviewer #1

(Remarks to the Author)

Reviewer #2

(Remarks to the Author)

The authors have addressed most of our comments and concerns. The requested additional discussion of related flavivirus proteases and clarification on obtained binding/inhibition data have been included and make the argument of this manuscript stronger. NMR data, while not disclosed here but alluded to with respect to future publications, may validate crystallographic findings. If this data is available for the compounds investigated in this manuscript, it should be included. However, if this data is only available for more advanced compounds that will be published separately, its inclusion might broaden the scope of this study too much.

Reviewer #3

(Remarks to the Author)

The authors have revised the manuscript which improved significantly.

Here are additional comments:

A more specific and precise title? eg. Discovery of Allosteric Inhibitors of the Zika Virus NS2B-NS3 Protease with Oral Efficacy in Mouse Models

The introduction is rather brief. The readers may wonder about the status of anti-Zika virus antiviral development, or anti-flavivirus in general.

Since the authors said, "We thank the referee for the suggestion, and we are currently evaluating the antiviral efficacy of IRBM-Z-2 against additional ZIKV strains" The data will be very useful to strengthen the claims in this manuscript and should be included in the revised manuscript.

The AG129 mouse model is a well-established model for ZIKV in vivo study. ZIKV infection in this strain results in a lethal outcome. Hence, use of the model for the study seems reasonable.

1) Fig4. The treatment regimen needs to be clarified. For the PO study, was a dose of 50 mg/kg twice a day given for each dosing, 100 mg/kg/day? Then it should be noted as 50 mg/kg, BID.

2) Viral loads in the brains should be presented for better evaluation of the efficacy.

3) Dosing for 14-days is not typical and no justifications are provided. A typical antiviral efficacy study uses 4-9 days of dosing and monitors if the treatment completely inhibited (thus wiped out the populations).

4) Based on the viral load in the blood (vRNAmia), the virus inoculum in the IRBM treated mice did not have a chance to initiate its replication. This is not clinically relevant. For better evaluation, a delayed treatment (DPI 2-3 for prior to neuroinvasion, or DPI4 for post-neuroinvasion) might be needed.

Reviewer #4

(Remarks to the Author)

Reviewer #5

(Remarks to the Author)

The authors identified a novel ZIKV NS2B-NS3 protease inhibitor through screening using a replicon assay. They identified the antiviral target of the molecule and performed a hit-optimization program which resulted in IRBM-Z-2. This last molecule showed sufficient potency and good pharmacokinetics allowing animal efficacy studies, and these demonstrate that IRBM-Z-2 can inhibit ZIKV replication in vivo and increase survival. The molecules are well described, the biochemical and cell-based activities on ZIKV and WNV are clear and the mechanism (although not entirely novel) is well supported through structural biology.

Main comment:

- A more specific title would be appropriate. For example: "A novel allosteric inhibitor of the Zikavirus NS2B-NS3 protease."
- There is no resistance selection experiment described for compound IRBM-Z-2. The fact that the resistance profile of IRBM-Z-2 has not been studied/characterized should be clearly mentioned as a limitation of the study.
- In line 328 it is stated "Remarkably, I156T was the only mutation identified, ... This suggest that other mutations are not-functional or do not confer resistance". We do not agree. Based on how the selection experiments were performed, it is possible that I156T is very easily selected for. It does not suggest that other resistance pathways are unlikely. This needs further attention.
- In the methods a ZIKV/Vero assay is described for IRBM-Z-1 but it is unclear where the results are shown. They are not in Fig1 nor in Table1. Only in figure 3D there are results from a full replication competent ZIKV CPE assay but then only for IRBM-Z-2. Similarly in line 70 "replication inhibition assay". "Replication" may likely be "replicon"? Please refer to all replicon assays clearly as "replicon" assays and not as "replication inhibition assay".
- In line 219 the authors claim that IRBM-Z-2 blocks the DENV2 protease, but there is no inhibition of the DENV2 replicon. How can this be explained? Any claim that IRBM-Z-2 may also inhibit DENV is not substantiated. There is no data of DENV2 in a replication assays and/or in animal models. Also further to line 355-356, it should be added that IRBM-Z-2 was not tested in a system with fully replication competent DENV, so its potential for DENV is not clear. This can also be mentioned as a limitation of this study.

Minor Comments

- In Fig1 b there are some results from the protease assay indicated as "EC50" which should be replaced with "IC50". Same in line 101. As the authors state in the legend of Table1, IC50 is used for biochemical and EC50 for cell-based assays. Perhaps it is not needed to explicitly write this in the legend of Table1.
- Line 70 "no cytotoxicity", please indicate max concentration measured. (same line 210)
- Indicate in legend fig 1 which cell lines were used to generate replicons.
- Line 130, refer to fig 1e for the SPR results
- Line 137: "X-ray structure" -> "Crystal structure"
- Line 147 "corresponding" -> "other"
- Line 280: "X-ray information" -> "structural biology information"
- Line 262: "infected" -> "in"
- Y-axis of fig 3d and e: "% Antiviral Activity" is perhaps more understandable for the reader then "% Reduction of CPE"

Version 2:

Reviewer comments:

Reviewer #1

(Remarks to the Author)

Reviewer #2

(Remarks to the Author)

All my comments have been addressed.

Reviewer #3

(Remarks to the Author)

No further comments. I recommend the paper to be published.

Reviewer #4

(Remarks to the Author)

Reviewer #5

(Remarks to the Author)

The authors revised the manuscript according to our suggestions. Last minor comment. In the conclusions the authors make the claim

(i) "To our knowledge, no current pre-clinical or clinical candidate exhibits comparable in vivo efficacy, making IRBM-Z-2 the most advanced, antiviral approach with demonstrated efficacy against ZIKV infection to date".

Comparing antiviral efficacies between different animal models is challenging as activity depends on multiple experimental factors (e.g. inoculum, end-point, ...).

(ii) "IRBM-Z-2 is the most advanced",, this should not be phrased as a certainty since pre-clinical stages of molecules are difficult to compare. We propose that the authors rephrase this sentence for example to "This in vivo efficacy positions IRBM-Z-2 among the more effective antiviral candidates currently under investigation." or "... among the more advanced ..."

Point-by-point response to the referees:

Referee #2 comments and response from authors

The authors have addressed most of our comments and concerns. The requested additional discussion of related flavivirus proteases and clarification on obtained binding/inhibition data have been included and make the argument of this manuscript stronger. NMR data, while not disclosed here but alluded to with respect to future publications, may validate crystallographic findings. If this data is available for the compounds investigated in this manuscript, it should be included. However, if this data is only available for more advanced compounds that will be published separately, its inclusion might broaden the scope of this study too much.

We thank the referees for their thoughtful considerations on our revised manuscript. As noted in your comment and in our previous point-by-point response, NMR studies of the binding of our allosteric inhibitors are currently being conducted using more advanced compounds. These experiments fully support the findings reported in the submitted manuscript and will be presented and discussed in detail in future publications focusing on these analogs.

Referee #3 comments and response from authors

The authors have revised the manuscript which improved significantly.

Here are additional comments:

A more specific and precise title? eg. Discovery of Allosteric Inhibitors of the Zika Virus NS2B-NS3 Protease with Oral Efficacy in Mouse Models

As advised by Referees #3 and #5 we have revised the manuscript title to “A novel allosteric inhibitor of the Zika virus NS2B-NS3 protease with oral efficacy in mouse models”.

The introduction is rather brief. The readers may wonder about the status of anti-Zika virus antiviral development, or anti-flavivirus in general.

We acknowledge the suggestion of the Referees and have expanded the Introduction to include a statement on the current status of flavivirus antiviral development. Specifically, we added the following sentence:

“Development of ZIKV antivirals is still at an early stage, with several experimental compounds reported to target different aspects of the viral lifecycle. These include SBI-0090799, which inhibits replication compartment formation; SYC-1307, an allosteric inhibitor of the NS2B-NS3 protease; and Galidesivir, a C-nucleoside analog that inhibits the viral RNA-dependent RNA polymerase (RdRp)⁵.”

Since the authors said, “We thank the referee for the suggestion, and we are currently evaluating the antiviral efficacy of IRBM-Z-2 against additional ZIKV strains” The data will be very useful to strengthen the claims in this manuscript and should be included in the revised manuscript.

As anticipated in our previous point-by-point letter, we have performed the antiviral efficacy assay of IRBM-Z-2 against two additional ZIKV strains, Padova and H/PF/2013. The results obtained are consistent with those observed for the PRVABC59 strain and are presented and discussed in the revised manuscript (see main text, page 8 and Supporting Information Fig. 13). The experimental details of the assay are described in the Methods section at page 19.

The AG129 mouse model is a well-established model for ZIKV in vivo study. ZIKV infection in this strain results in a lethal outcome. Hence, use of the model for the study seems reasonable.

1) Fig4. The treatment regimen needs to be clarified. For the PO study, was a dose of 50 mg/kg twice a day given for each dosing, 100 mg/kg/day? Then it should be noted as 50 mg/kg, BID.

We thank the reviewers for their observation and have revised the figure 4 legend on page 11 accordingly. For the PO efficacy study, the administration regimen consisted of 100 mg/kg twice daily (BID), corresponding to a total daily dose of 200 mg/kg/day.

2) Viral loads in the brains should be presented for better evaluation of the efficacy.

In this study, we evaluated viral loads in blood, rather than brain, as our primary objective was to assess the prophylactic potential of IRBM-Z-2 for ZIKV, particularly considering the challenges and ethical constraints associated with conducting clinical trials in pregnant women. At present, we are expanding our research to investigate the efficacy of more advanced compounds in neuroinvasive illnesses caused by other orthoflavivirus members, with a specific focus on their impact on the central nervous system. Accordingly, viral loads in brain tissue are being measured in ongoing *in vivo* studies with a more advanced compound. The outcomes of these experiments will be presented in future publications on this compound and its analogs.

3) Dosing for 14-days is not typical and no justifications are provided. A typical antiviral efficacy study uses 4-9 days of dosing and monitors if the treatment completely inhibited (thus wiped out the populations).

The study duration was extended to 14 days to enable a comprehensive evaluation of the compound's efficacy in suppressing ZIKV replication and to assess its safety during prolonged treatment, thereby providing a more accurate representation of its therapeutic potential. This rationale has been incorporated on page 10 of the manuscript to clarify our study design.

4) Based on the viral load in the blood (vRNA_m), the virus inoculum in the IRBM treated mice did not have a chance to initiate its replication. This is not clinically relevant. For better evaluation, a delayed treatment (DPI 2-3 for prior to neuroinvasion, or DPI4 for post-neuroinvasion) might be needed.

We acknowledge the point of the referee regarding the clinical relevance of the results

presented in this manuscript. However, following the excellent outcome of the proof-of-concept (PoC) study, our goal was to take a step further and design a second *in vivo* experiment to evaluate the prophylactic potential of the compound in a clinically relevant setting. We are currently investigating the efficacy of more advanced compounds in neuroinvasive disease models, which will provide a more comprehensive and clinically relevant assessment of our inhibitors.

Referee #4 and #5 comments and response from authors

The authors identified a novel ZIKV NS2B-NS3 protease inhibitor through screening using a replicon assay. They identified the antiviral target of the molecule and performed a hit-optimization program which resulted in IRBM-Z-2. This last molecule showed sufficient potency and good pharmacokinetics allowing animal efficacy studies, and these demonstrate that IRBM-Z-2 can inhibit ZIKV replication in vivo and increase survival. The molecules are well described, the biochemical and cell-based activities on ZIKV and WNV are clear and the mechanism (although not entirely novel) is well supported through structural biology.

Main comment:

- A more specific title would be appropriate. For example: “A novel allosteric inhibitor of the Zikavirus NS2B-NS3 protease.”

We have revised the manuscript title to “A novel allosteric inhibitor of the Zika virus NS2B-NS3 protease with oral efficacy in mouse models”,

- There is no resistance selection experiment described for compound IRBM-Z-2. The fact that the resistance profile of IRBM-Z-2 has not been studied/characterized should be clearly mentioned as a limitation of the study.

In the description of the biological profile of IRBM-Z-2, at page 8, we comment “IRBM-Z-2 lost nearly two orders of magnitude of activity when tested against the I156T NS2B-NS3 mutant, in both biochemical and replicon assays, consistent with what was observed for IRBM-Z-1”, those data being shown in Table 1. We believe that these results sufficiently prove that NS2B-NS3 protease is the biological target of IRBM-Z-2 and no resistance selection experiment is needed. Furthermore, in our manuscript, we present the results of kinetic and binding studies, together with the crystal structure of the ZIKV NS2B-NS3 protease in complex with IRBM-Z-2. In our opinion, these comprehensive data confirm that the mechanism of action of IRBM-Z-2 is the same as that of IRBM-Z-1.

- In line 328 it is stated “Remarkably, I156T was the only mutation identified, ... This suggest that other mutations are not-functional or do not confer resistance”. We do not agree. Based on how the selection experiments were performed, it is possible that I156T is very easily selected for. It does not suggest that other resistance pathways are unlikely. This needs further attention.

We acknowledge the referee’s point and, to avoid any misunderstanding, we have rephrased the sentence in page 12 as follows:

“Remarkably, I156T was the only mutation identified in the resistance-selection experiments.

This finding suggests that the mutation does not significantly impair replication and is the most readily selected among potential variants under our experimental conditions.”

- In the methods a ZIKV/Vero assay is described for IRBM-Z-1 but it is unclear where the results are shown. They are not in Fig1 nor in Table1. Only in figure 3D there are results from a full replication competent ZIKV CPE assay but then only for IRBM-Z-2. Similarly in line 70 “replication inhibition assay”. “Replication” may likely be “replicon”? Please refer to all replicon assays clearly as “replicon” assays and not as “replication inhibition assay”.

We appreciated the referee’s inquiry and included a phrase disclosing the result obtained for IRBM-Z-1 in the ZIKV anti-infectivity assay (EC₅₀: 6.25 μM) on page 3. In addition, in line 70, we have replaced the term “*replication inhibition assay*” with “*replicon assay*” for greater accuracy.

- In line 219 the authors claim that IRBM-Z-2 blocks the DENV2 protease, but there is no inhibition of the DENV2 replicon. How can this be explained? Any claim that IRBM-Z-2 may also inhibit DENV is not substantiated. There is no data of DENV2 in a replication assay and/or in animal models. Also further to line 355-356, it should be added that IRBM-Z-2 was not tested in a system with fully replication competent DENV, so its potential for DENV is not clear. This can also be mentioned as a limitation of this study.

We acknowledge the concern of the referee regarding the antiviral activity of IRBM-Z-2 against DENV. We would like to emphasize that IRBM-Z-2 inhibits DENV2 NS2B-NS3 protease with an IC₅₀ of 2.1 μM, which provides approximately a one-log window relative to the concentration limit of the cell-based assay, a difference that is not uncommon in antiviral studies. Moreover, Although the exact reason remains unclear, it can be hypothesized that the inhibition of the NS2B-NS3 protease has a stronger impact on replication in the ZIKV replicon system than in the DENV2 replicon.

For this reason, and due to the lack of detectable activity in the DENV2 replicon, we did not proceed with animal model studies of DENV2.

However, as suggested by the referee, to address this concern, we have added the statement “even though the potential for DENV needs to be even though the potential efficacy against DENV needs to be evaluated further.” at page 12 of the manuscript main text.

In addition, in follow-up work on this series, we have developed analogs of IRBM-Z-2 with nanomolar potency against the DENV2 NS2B-NS3 protease and submicromolar activity in both replicon and anti-infectivity assays. These results will be presented in future publications.

Minor Comments:

- In Fig1 b there are some results from the protease assay indicated as “EC50” which should be replaced with “IC50”. Same in line 101. As the authors state in the legend of Table1, IC50 is used for biochemical and EC50 for cell-based assays. Perhaps it is not needed to explicitly write this in the legend of Table 1.

Done. In Fig. 1c, the parameter EC₅₀ of the ZIKV NS2B-NS3 protease assay has been corrected to IC₅₀. Similarly, in Fig. 1d, the assay parameter EC₅₀ of all four biochemical assays has been replaced by IC₅₀. The legend of Figure 1 has been updated accordingly. Additionally, the definitions of IC₅₀, EC₅₀ and CC₅₀ have been removed from the legend of Table 1, and the

wording of the sentence referring to the cell viability assay has been revised for clarity.

- *Line 70 “no cytotoxicity”, please indicate max concentration measured. (same line 210)*

Done. In lines 70 and 210, to clarify the maximum concentration of the cell viability assay, we have added the sentence: “up to 32 μ M concentration”.

- *Indicate in legend fig 1 which cell lines were used to generate replicons.*

Done. We have added the phrase “Replicon assays were carried out using Vero cells” in the legend of Figure 1 to indicate the cell line used in all the cell-based assays and to generate replicons.

- *Line 130, refer to fig 1e for the SPR results*

Done. We have added the sentence “(Fig. 1e)” in brackets to the SPR kinetic studies results.

- *Line 137: “X-ray structure” -> “Crystal structure”*

Done. We have replaced the expression “X-ray structure” by “Crystal structure”.

- *Line 147 “corresponding” -> “other”*

Done. We have replaced the word “corresponding” by “other”.

- *Line 280: “X-ray information” -> “structural biology information”*

Done. We have replaced the expression “X-ray information” by “structural biology information”.

- *Line 262: “infected” -> “in”*

Done. We have deleted the word “in” in the legend of Figure 3.

- *Y-axis of fig 3d and e: “% Antiviral Activity” is perhaps more understandable for the reader than “% Reduction of CPE”*

Done. We have replaced the expression “% Reduction of CPE” by “% Antiviral Activity”.

Referee #5 comments and response from authors

The authors revised the manuscript according to our suggestions. Last minor comment. In conclusion, the authors make the claim:

(i) "To our knowledge, no current pre-clinical or clinical candidate exhibits comparable in vivo efficacy, making IRBM-Z-2 the most advanced, antiviral approach with demonstrated efficacy against ZIKV infection to date"..

Comparing antiviral efficacies between different animal models is challenging as activity depends on multiple experimental factors (e.g. inoculum, end-point, ...).

(ii) "IRBM-Z-2 is the most advanced", this should not be phrased as a certainty since pre-clinical stages of molecules are difficult to compare. We propose that the authors rephrase this sentence for example to "This in vivo efficacy positions IRBM-Z-2 among the more effective antiviral candidates currently under investigation." or "... among the more advanced ..."

We acknowledge the referee's point, and we have rephrased the sentence on page 12 as follows:

"This robust *in vivo* efficacy positions IRBM-Z-2 among the more effective pre-clinical or clinical antiviral candidates currently under investigation."